# A Comparative Analysis of Different *Xenorhabdus* Strains Reveals a Virulent Factor, Cyclic Pro-Phe, Using a Differential Expression Profile Analysis of Non-Ribosomal Peptide Synthetases

**DOI:** 10.3390/insects15090710

**Published:** 2024-09-17

**Authors:** Gahyeon Jin, Falguni Khan, Yonggyun Kim

**Affiliations:** Department of Plant Medicals, Andong National University, Andong 36729, Republic of Korea; gsh07129@daum.net (G.J.); falguni.agri@gmail.com (F.K.)

**Keywords:** entomopathogenic bacteria, *Xenorhabdus*, non-ribosomal peptide synthetase, immunity, secondary metabolite

## Abstract

**Simple Summary:**

*Xenorhabdus* is a bacterial genus that symbioses with entomopathogenic nematodes, specifically classified within *Steinernema*. These nematodes transport the bacteria into target insects, where the bacteria suppress the host’s immune responses and induce septicemia. Within the cadaver of the insect, the nematodes reproduce and re-establish their association with the symbiotic bacteria. This study identifies significant variability in insecticidal virulence across different *Xenorhabdus* species or strains. This variability arises from differences in their growth rates and suppressive activities against host immune functions. Analysis of secondary metabolites revealed the presence of several peptides in the virulent strains. Further analysis of non-ribosomal peptide synthetase genes suggested cyclo Pro-Phe (cPF) as a virulence factor, which suppressed insect immune responses and enhanced the bacterial virulence.

**Abstract:**

Entomopathogenic bacteria, classified into the genus *Xenorhabdus*, exhibit a dual lifestyle as mutualistic symbionts to *Steinernema* nematodes and as pathogens to a broad range of insects. Bacterial virulence depends on toxin proteins that induce toxemia and various immunosuppressive secondary metabolites that cause septicemia. Particularly, the immunosuppressive properties of *Xenorhabdus* bacteria determine the variability of their insecticidal activities. This study explored the role of peptide metabolites in virulence and its variation among six bacterial strains across three species: *X. nematophila*, *X. bovienii*, and *X. hominickii*. Initially, their virulence significantly varied against a susceptible lepidopteran host, *Maruca vitrata*, but showed less variation against a tolerant coleopteran host, *Tenebrio molitor*, with high median lethal bacterial doses. In *M. vitrata*, virulence was strongly correlated with bacterial growth rate and inhibitory activity against phospholipase A_2_. Secondly, the six strains differed in the compositions of their secreted secondary metabolites, analyzed by GC-MS following ethyl acetate extraction. Notably, there was significant variation in the production of di- or tetra-peptides. Highly virulent strains commonly produced the cyclic Pro-Phe (cPF). Thirdly, the expression of non-ribosomal peptide synthetase (*NRPS*) genes varied greatly among the strains. NRPS genes were minimally expressed in the tolerant *T. molitor* and highly expressed in the susceptible *M. vitrata*. In *M. vitrata*, specific *NRPS* genes were markedly expressed in the virulent strains. Finally, cPF demonstrated potent immunosuppressive activity against the cellular and humoral responses of *M. vitrata*. The addition of cPF significantly enhanced the virulence against the tolerant *T. molitor*. These findings suggest that immunosuppression is necessary for the pathogenicity of *Xenorhabdus* bacteria, wherein NRPS products play a critical role in suppressing immune-associated factors in target insects.

## 1. Introduction

Two bacterial genera, *Xenorhabdus* and *Photorhabdus*, are in mutual symbiosis with *Steinernema* and *Heterorhabditis* nematodes, respectively [1]. These bacteria inhabit a specialized gut luminal pouch known as the receptacle in the host nematodes during the infective juvenile (IJs) stage [2]. In the soil, the nematodes locate host insects through behaviors classified as ambushing (waiting for prey) or cruising (actively searching for prey) [3]. Upon entry into the insect hemocoel via natural openings, IJs expel their symbiotic bacteria by regurgitation [4]. In reaction to bacterial invasion, the targeted insects activate and induce immune defenses [5]. As eicosanoids mediate both cellular and humoral immune responses, their biosynthetic pathways are typically activated immediately after the pathogen recognition signal [6]. To counter these immune responses, the bacteria secrete secondary metabolites that suppress or directly inhibit acute immune reactions [7]. Phospholipase A_2_ (PLA_2_) is the committed step in eicosanoid biosynthesis and serves as a molecular target for the secondary metabolites from *Xenorhabdus* and *Photorhabdus* [8,9].

*Xenorhabdus* and *Photorhabdus* exhibit variability in their insecticidal activities against the same insect host and among different insect species [7]. Specifically, variations in insecticidal activity have been observed among different strains of a particular bacterial species [7]. Several hypotheses have been proposed to explain the variation in bacterial virulence of *Xenorhabdus* and *Photorhabdus*. One hypothesis, derived from a comparative genomic analysis of *X. bovienii* strains, suggests the presence of gene clusters in virulent strains, including the type VI secretion system (T6SS), which may be crucial for competing with other microbiota in the target insect hosts [10]. Another hypothesis relates to the bacterial flagella mobility of *Xenorhabdus* and *Photorhabdus*, asserting that their swimming and swarming motility impacts insect colonization, and that the flagellar type 3 secretion system serves as an export apparatus for virulence proteins [11]. This hypothesis extends the control of virulence factors to include a transcriptional regulon, Lrh1, regulated by a global transcription regulator, leucine-responsive regulatory protein (Lrp). The expression levels of *Lrp* regulate various downstream regulons, affecting the expression of pathogenic genes related to secondary metabolites, including non-ribosomal peptide synthetase (*NRPS*) genes [12]. In *X. nematophila*, low levels of *Lrp* expression induce bacterial virulence and host immunosuppression, whereas high levels reduce bacterial virulence [13]. Additionally, other transcriptional regulons like LeuO and HexA contribute to the production of bacterial secondary metabolites in *Xenorhabdus* and *Photorhabdus* [14]. The importance of these regulons is underscored by their regulation through small regulatory RNAs, where *ArcZ* (=Hfq-dependent sRNA) is complementary to *HexA* mRNA and controls approximately 15% of all transcripts in *Photorhabdus* and *Xenorhabdus* [15].

Immunosuppressive activity and toxemia contribute to the insecticidal virulence of *Xenorhabdus* and *Photorhabdus* [16]. Variation in inhibitory activity against PLA_2_ catalytic activity is closely correlated with bacterial virulence, as suppressed enzyme activity impairs the production of immune-mediating eicosanoids [17]. The PLA_2_ inhibitors derived from *Xenorhabdus* and *Photorhabdus* were isolated from the bacterial culture broth following ethyl acetate extraction and include seven compounds featuring a common phenylethyl motif [9]. Mollah et al. [7] postulated that differential *NRPS* expression leads to variations in virulence between two strains of *X. hominickii* through immunosuppression. This implies that NRPS-catalyzed products inhibit PLA_2_, thereby shutting down eicosanoid biosynthesis and leading to fatal immunosuppression.

This study aimed to test the hypothesis that *NRPS* expression in bacteria plays a critical role in determining virulence against target insects. Consequently, it compared six different strains of three *Xenorhabdus* species regarding their virulence against two insect targets: a lepidopteran *Maruca vitrata* and a coleopteran *Tenebrio molitor*, which have been demonstrated in their immune responses controlled by eicosanoids [7]. Secondary metabolites from the six strains were analyzed using GC-MS on ethyl acetate extracts from the bacterial culture broth containing PLA_2_ inhibitors [9]. *NRPS* genes, predicted from the bacterial genomes, were analyzed for their expression levels in the target insects. From this analysis, an NRPS product specific to virulent strains was identified, which exhibited inhibitory activity against PLA_2_ to suppress target insect immunity.

## 2. Materials and Methods

### 2.1. Insect Rearing

Larvae of *M. vitrata* were collected from adzuki bean (*Vigna angularis*) fields in Suwon, Republic of Korea. They were subsequently cultured on an artificial diet as outlined by Jung et al. [18]. Adults were provided with 10% sucrose. Larvae and adults of *T. molitor* were obtained from Bio Utility, Inc. (Andong, Republic of Korea) and reared on a wheat bran-based diet supplemented with cabbage, following the methodology detailed by Liu et al. [19]. Rearing conditions were 25 ± 2 °C temperature, 16:8 h (L:D) photoperiod, and 60 ± 10% relative humidity.

### 2.2. Bacterial Culture

Two strains (Xh-ANU and Xh-DY) of *X. hominickii* were sourced from laboratory culture stock [20]. *X. bovienii* strain SS2004 (Xb-SS2004) was provided by Dr. Sophie Gaudriault from the University of Montpellier, France. *X. bovienii* strain ANU (Xb-ANU) was isolated from *S. feltiae* [21]. Two strains (Xn-FK and Xn-GH) of *X. nematophila* were derived from different field populations of *S. carpocapsae* collected in Andong, Republic of Korea (Appendix A). These bacteria were cultured in tryptic soy broth (TSB: Difco, Sparks, MD, USA) at 28 °C [20].

### 2.3. Chemicals

Phospholipase A_2_ (PLA_2_) enzyme assay kits were obtained from Cayman Chemical (Ann Arbor, MI, USA). 4′,6-Diamidine-2′-phenylindole dihydrochloride (DAPI) and 3-(4,5-dimethylthiazole-2-yl)-2,5-diphenyl tetrazolium bromide (MTT) were acquired from Sigma-Aldrich Korea (Seoul, Republic of Korea). Alexa Fluor 488, used as a dye analogous to fluorescein isothiocyanate (FITC), was procured from Thermo Fisher Scientific (Waltham, MA, USA). For RNA preparation, diethyl pyrocarbonate (DEPC) water was prepared by mixing 1 mL of DEPC with 1 L of deionized water, incubated for 12 h at 25 °C, autoclaved twice, and stored at room temperature until use. Phosphate-buffered saline (PBS, pH 7.4) consisted of 100 mM phosphate and 0.7% NaCl. The anticoagulant buffer (ACB, pH 4.5) was composed of 186 mM NaCl, 17 mM Na2EDTA, and 41 mM citric acid.

### 2.4. Bioinformatics Analysis

The whole genome sequences of *X. hominickii*, *X. nematophila*, and *X. bovienii* were obtained from GenBank with accession numbers NZ_CP016176.1, NZ_CP060401.1, and NZ_FO818637.1, respectively. *NRPS* genes were predicted using NCBI gene annotation. Functional modules and products of each NRPS were determined using the antiSMASH bacterial version software (antismash.secondarymetabolites.org/, accessed on 1 September 2024). Synteny analysis of NRPS genes was performed using the CLC workbench Whole Genome Alignment Plugin Comparative Analysis.

### 2.5. Bioassay on Insecticidal Activity of the Bacteria

Third instar (L3) larvae of *M. vitrata* and L8 larvae of *T. molitor* were used. Freshly cultured bacteria were injected into each larva with 1 μL volume containing 1.5 × 10^3^ colony-forming units (CFU) using a micro-syringe. The treated larvae were kept at the rearing conditions with fresh diets. Mortality was assessed 72 h post-injection (pi) as most dead insects were observed at 48 h pi. Each treatment included 10 larvae and was replicated three times with different cohorts. The control used an injection of PBS used for resuspending the test bacteria.

### 2.6. Measurement of Bacterial Growth Curve

To monitor bacterial growth in *M. vitrata* and *T. molitor*, 2 μL (2 × 10^7^ CFU/mL) of freshly cultured bacteria from six different strains was injected into the hemocoel of each larva using a micro-syringe (Hamilton, Reno, NE, USA). In this assay, we used the last instar larvae of *M. vitrata* and the matured larvae (>10 mm body length) of *T. molitor*. Considering the larval hemocoel volume of 200 μL [22], we inoculated 2 × 10^8^ CFU of bacteria into 1 L of TSB medium. After various incubation periods, we collected the culture media and hemolymph and spread them onto tryptic soy agar plates. Following 16 h of incubation at 28 °C, we counted the colony numbers to monitor bacterial growth.

### 2.7. Expression Analysis of Each NRPS Gene

To analyze bacterial gene expressions in TSB culture, we centrifuged the broth to obtain cell pellets. Each larva served as an experimental unit for expression analysis in infected larvae, as detailed earlier. Following the manufacturer’s instructions, we treated the bacterial pellet or insect sample with Trizol reagent (Invitrogen, Carlsbad, CA, USA). We then processed the resulting mixture according to the recommended procedure and resuspended the extracted RNA from each sample in 50 μL of DEPC water. We determined its concentration using a spectrophotometer (Nanodrop, Thermo Fisher Scientific, Wilmington, DE, USA). For cDNA synthesis, we used 400 ng of RNA and an RT premix (Intron Biotechnology, Seoul, Republic of Korea) containing random primers.

For RT-PCR, we amplified cDNAs using gene-specific primers (Appendix A) and Taq polymerase (GeneALL, Seoul, Republic of Korea). The PCR reaction commenced with an initial denaturation at 95 °C for 5 min, followed by 35 amplification cycles consisting of denaturation at 95 °C for 1 min, annealing at 53–58 °C for 1 min, and extension at 72 °C for 1 min. It concluded with a final extension at 72 °C for 10 min. The presence of the amplified product was verified by analyzing the PCR product through 1% agarose gel electrophoresis.

For RT-qPCR, we used the Power Syber Green PCR Master Mix (Toyobo, Osaka, Japan) along with gene-specific primers following the guidelines established by Bustin et al. [23]. The PCR reaction began with an initial heat treatment at 95 °C for 30 s, followed by annealing at 53~58 °C for 30 s, and extension at 72 °C for 30 s. To normalize the expression level in each qPCR sample, we used the reference gene, 16S rRNA, and its specific primers (Appendix A). We conducted quantitative analysis using the comparative CT method [24] and ensured statistical reliability by independently replicating all experiments three times.

### 2.8. Extraction of Bacteria Secondary Metabolites

The test bacteria were cultured in 1 L of tryptic soy broth for 72 h at 28 °C with continuous agitation at 180 rpm. After the incubation, the bacterial culture broth was centrifuged at 12,500× *g* for 20 min at 4 °C; the resulting supernatant was mixed with an equal volume of ethyl acetate (1 L). The organic phase was separated using a separate funnel. The aqueous phase underwent two additional extractions with ethyl acetate, using the earlier described extraction procedure. The combined 3 L of organic phase extract was then concentrated using a rotary evaporator (N-1110 Eyela, Tokyo, Japan) at 30 °C. The resulting dried pellet containing metabolites was weighed and resuspended in DMSO to a concentration of 100 ppm. The compounds in the ethyl acetate extracts were identified through GC-MS analysis. The GC system (7890B, Agilent Technologies, Santa Clara, CA, USA) utilized an MS unit (5977A Network, Agilent Technologies). An HP5 MS column (Agilent Technologies), a non-polar column with an internal diameter of 30 m × 250 μm and a film thickness of 0.25 μm, was employed. Helium served as the carrier gas at a flow rate of 1 mL/min. The injector temperature was set at 200 °C in split mode, with a split ratio of 10:1. The initial oven temperature started at 100 °C, was held for 3 min, then increased to 300 °C at a rate of 5 °C/min. The final oven temperature was maintained at 300 °C for an additional 10 min, totaling a running time of 53 min. Mass spectra were acquired in electron ionization (EI) mode at 70 eV, covering a scanning range of 33–550 *m*/*z*. For compound identification, the purified samples were compared against mass spectra from the NIST11 database (U.S. Department of Commerce, Gaithersburg, MD, USA) and relevant literature, available at http://nistmasspeclibrary.com (accessed on 1 September 2024).

### 2.9. PLA_2_ Activity Assay

PLA_2_ activities were quantified using the sPLA_2_ assay kit from Cayman Chemical, which included a diheptanoyl thio-phosphatidylcholine substrate. The assessment followed the method outlined in Vatanparast et al. [25]. For the experiment assessing inhibition of PLA_2_ enzyme activity, fat bodies of each insect were isolated and homogenized in PBS.

### 2.10. Phenoloxidase (PO) Enzyme Assay

Hemolymph was collected from the test larvae used for bioassay and subsequently divided into hemocytes and plasma, adhering to the outlined protocol. To measure PO activity in the plasma, we used L-3,4-dihydroxyphenylalanine (DOPA) as a substrate. A reaction mixture of 200 μL was prepared, consisting of 10 μL of plasma, 10 μL of DOPA, 2 μL of bacterial metabolites (100 ppm), and 178 μL of PBS. The absorbance (ABS) of the reaction mixture was measured at 495 nm using a VICTOR multi-label plate reader (PerkinElmer, Waltham, MA, USA). PO activity was calculated as ABS/min/mL. Each treatment was repeated in triplicate.

### 2.11. Statistical Analysis

All studies were conducted using one-way ANOVA with PROC GLM of the SAS program [26]. Mortality data were subjected to arcsine transformation and analyzed by ANOVA. Means were compared using the least squared difference (LSD) test. All experiments included three biologically independent replicates, and data were plotted as mean ± standard error using Sigma plot v10.0 (Grafiti LLC, Palo Alto, CA, USA). The median lethal dose (LD_50_), median inhibitory concentration (IC_50_), and correlation analyses were performed using GraphPad Prism 9 (GraphPad Software, Boston, MA, USA). The bacterial growth rate (*r*) was estimated using nonlinear and multiple regression analyses in GraphPad Prism 9.

## 3. Results

### 3.1. Variation in Insecticidal Virulence of Six Different Xenorhabdus Strains

Each of three *Xenorhabdus* species (*X. nematophila*, *X. hominickii*, and *X. bovienii*) had two strains. These six strains were compared for their insecticidal activities against lepidopteran (*M. vitrata*) and coleopteran (*T. molitor*) species (Figure 1). A clear dose-dependent virulence was observed in *M. vitrata* (*F* = 370.8; df = 5, 60; *p* < 0.0001) and *T. molitor* (*F* = 121.08; df = 5, 60; *p* < 0.0001) (Figure 1A). In insects, *T. molitor* was more tolerant (11.8~96.9 fold in LD_50_) than *M. vitrata* to each bacterial strain (Figure 1B). In *M. vitrata*, a virulence variation of 1.34- ~ 10.42-fold in the median lethal dose was recorded. However, the virulence variation observed against *T. molitor* ranged from 1.02- to 1.40-fold.

### 3.2. Correlation of the Bacterial Biological Characters with Their Insecticidal Virulence

The six *Xenorhabdus* strains were cultured under three different conditions: TSB and two different insect hosts (Figure 2A). In the TSB medium, all six strains sustained the lag phase for 12 h pi, progressed to the log phase until 42 h pi, and then entered the stationary phase. There was minimal variation (*F* = 3.32; df = 5, 108; *p* = 0.0078) in the growth rates of the bacterial strains in TSB medium. Conversely, the bacterial populations in the insect host, *M. vitrata*, grew rapidly, reaching a stationary phase by 24 h post-inoculation. Significant differences (*F* = 129.53; df = 5, 96; *p* < 0.0001) were observed among the growth rates of these bacterial strains. In *T. molitor*, the bacteria also exhibited rapid growth similar to that in *M. vitrata*, with relatively minor variations (*F* = 18.71; df = 5, 96; *p* < 0.0001) compared to *M. vitrata* in terms of bacterial growth rates.

Different strains of *Xenorhabdus* bacteria were evaluated for their ability to suppress insect immunity by inhibiting eicosanoid biosynthesis. PLA_2_ catalyzes the key step in eicosanoid biosynthesis and serves as a molecular target for pathogenic *Xenorhabdus* bacteria [8]. Two PLA_2_s of secretory PLA_2_ (sPLA_2_) and intracellular PLA_2_ (iPLA_2_), are known to be involved in insect immune responses [27,28]. Organic extracts from the culture broth of six different *Xenorhabdus* spp. were obtained and assessed for their ability to inhibit PLA_2_ activities in two insect targets (Figure 2B). In *M. vitrata*, the bacterial extracts inhibited the enzyme activities of both sPLA_2_ and iPLA_2_ in a dose-dependent manner, with significant variation among the bacteria in the inhibition of PLA_2_ activities: *F* = 28.44; df = 5, 96; *p* < 0.0001 for sPLA_2_ and *F* = 122.14; df = 5, 96; *p* < 0.0001 for iPLA_2_. In *T. molitor*, the bacterial extracts also demonstrated dose-dependent inhibition of PLA_2_ activities but showed little variation in inhibitory activities among bacterial strains against sPLA_2_ (*F* = 0.16; df = 5, 96; *p* = 0.9749) and slight inhibitory activity against iPLA_2_ (*F* = 8.01; df = 5, 96; *p* < 0.0001).

These biological characteristics of the six bacterial strains were correlated with their insecticidal virulence (Figure 2C). In *M. vitrata*, insecticidal virulence was strongly correlated with growth rate and PLA_2_ inhibition intensity. However, these correlations were low and not significant in *T. molitor*.

### 3.3. Variation in the Secondary Metabolites Produced by Six Xenorhabdus Strains

GC-MS analysis was conducted on the bacterial organic extracts, revealing a total of 368 compounds (Figure 3A). Among the three species, *X. nematophila* produced a higher number of secondary metabolites compared to the other two species. Within each species, the two strains differed in their secondary metabolite profiles, with common compounds accounting for less than half (25~46%) of the total compounds produced per species (Figure 3B). Despite these variations in secondary metabolites among *Xenorhabdus* bacteria, 34 compounds were consistently present across the six species (Appendix A). *X. nematophila* exhibited a greater number of species-specific compounds than the other two species.

The secondary metabolites were categorized according to their chemical nature, including peptides, indoles, phenylamides, and hydrocarbons (Figure 4A). Although the quantities and types of these compounds varied among the six bacterial strains, peptides showed the most significant differences (Figure 4B). Among the six peptides, cyclic Pro-Phe (cPF) was exclusively found in virulent strains of the three bacterial species.

### 3.4. Variation of Non-Ribosomal Peptide Synthetase (NRPS) Genes among Six Xenorhabdus Strains

To explain the variation in peptide metabolite production, NRPS genes were analyzed in three *Xenorhabdus* species using genomic data (Figure 5A and Appendix A). *X. hominickii* possesses eight NRPS genes (*Xh-N1*~*Xh-N8*) within its approximately 4.5 Mb genome. *X. bovienii* and *X. nematophila* each have seven NRPS genes (*Xb-N1*~*Xb-N7* and *Xn-N1*~*Xn-N7*, respectively) in their approximate 4.3 Mb genomes. Synteny analysis revealed limited co-linearity among the NRPS genes across these species, with *X. nematophila* showing distinct divergence.

These NRPS genes were analyzed for their expression in two different insect hosts (Figure 5B). Interestingly, all three bacterial species exhibited relatively high expression levels in *M. vitrata* but not in *T. molitor*. Additionally, these NRPS genes demonstrated higher expression levels in the virulent strain of each bacterial species. For example, *Xh-N2* was highly expressed in the virulent ANU strain compared to the DY strain in *M. vitrata*. *Xn-N4* was highly expressed in the virulent FK strain compared to the GH strain in *M. vitrata*. *Xb-N1* and *Xb-N2* were highly expressed in the virulent SS-2004 strain compared to the ANU strain in *M. vitrata*.

### 3.5. A Dipeptide Metabolite, cPF, Specific to Virulent Xenorhabdus Strain

cPF (Figure 6A) was identified in the bacterial culture broth of *X. hominickii* and predicted to be produced by the Xh-N6 catalytic machinery [12]. It was also detected in our current assays. It was detected in the virulent strains of the three bacterial species. The virulent properties of cPF were analyzed by assessing its synthetic gene analysis (Figure 6B). *Xh-N6* was highly expressed in the virulent strain Xh-ANU compared to Xn-DY in *M. vitrata*. However, its expression did not differ in *T. molitor*. cPF inhibited PLA_2_ activities extracted from the insects, inhibiting PLA_2_s of *M. vitrata* more than those of *T. molitor* (Figure 6C). It also inhibited PO activities in the insects, with PO from *M. vitrata* being more susceptible than that of *T. molitor* (Figure 6D). The addition of cPF significantly enhanced the bacterial virulence (left panel in Figure 6E). The tolerance of *T. molitor* to the bacterial infection was reduced with the addition of cPF (right panel in Figure 6E).

## 4. Discussion

Secondary metabolites produced by fungi, plants, and bacteria are valuable for developing new drugs due to their unique structure, high activity, and selectivity [29]. Particularly, *Photorhabdus* and *Xenorhabdus* bacteria are considered valuable sources of these secondary metabolites because they encode several putative biosynthetic pathways for natural product biosynthesis, which are conserved across their ecological niches and fulfill vital ecological functions [30]. Additionally, along with the immunosuppressive activities of these bacterial metabolites, several bacteria within *Xenorhabdus* produce antiprotozoal compounds such as fabclavines, xenocoumacins, xenorhabdins, and PAX peptides against human pathogenic protozoa [31]. This study provides information concerning the secondary metabolites produced by various strains of *Xenorhabdus* bacteria in terms of compounds and biological activities to explain the virulence variation among different strains of *Xenorhabdus*.

Variation in bacterial virulence differed among insect hosts, notably *M. vitrata* was more susceptible to bacterial infection than *T. molitor*. This differential virulence was also observed among bacterial host nematodes. For instance, *S. feltiae* hosting *X. bovienii* exhibited a preference for *Spodoptera exigua* over *T. molitor*, likely due to less potent immune response of *S. exigua* compared to that of *T. molitor* [32]. Indeed, our current study revealed that various *Xenorhabdus* strains suppressed PLA_2_ activities in correlation with virulence against the susceptible host, *M. vitrata*. This supports the hypothesis that inhibiting PLA_2_ activity is essential for the bacterial virulence of *Xenorhabdus* and *Photorhabdus* [8,21]. PLA_2_ catalyzes the biosynthesis of various eicosanoids, mediating both cellular and humoral immune responses in insects [33]. Thus, the bacteria’s immunosuppressive effects on host insects are attributable to inhibitory actions against PLA_2_. Conversely, the inhibitory effects of the bacterial strains on PLA_2_ activity did not correlate with bacterial virulence in *T. molitor*, implying a minimal production of virulent factors by the bacteria in the more resistant insect host. In other words, host factors derived from *T. molitor* might inhibit the transcriptional signal leading to secondary metabolites responsible for blocking PLA_2_. Seo et al. [9] identified seven PLA_2_ inhibitors produced by *X. nematophila*, including those with a phenylethyl chemical backbone such as benzylideneacetone. GameXPeptide (GXP), a cyclic pentapeptide produced by both *Xenorhabdus* and *Photorhabdus*, stemming from the catalytic product of a specific NRPS, inhibits insect PLA_2_ [30,34]. The synthesis of GXP is likely regulated by several bacterial transcriptional factors influenced by the insect host factors [12,35]. This indicates the presence of antibacterial factors that disrupt bacterial signaling in *T. molitor* and warrants further investigation.

Interestingly, the bacterial virulence in the susceptible host, *M. vitrata*, was strongly correlated with their growth rate. *Xenorhabdus* bacteria experience a phase variation based on environmental conditions, where phase 1 bacteria are mobile/flagellated and produce antibiotics, hemolysins, immune suppressors, and toxins, whereas phase 2 bacteria in *X. nematophila* lack these characteristics and have a colony size approximately 10 times smaller [36,37]. Consequently, the infective juveniles of nematodes prefer phase 1 bacteria for their pathogenicity. This variation in phenotypes can be attributed to the gene expression of the functional leucine-responsive regulatory protein (Lrp) [38]. Although Lrp expression occurs at the stationary phase in TSB culture, it is markedly expressed in the insect host at the early exponential stage in *X. hominickii* [12]. Additionally, the over-expression of *Lrp* increases the production of virulent factors and enhances the immunosuppressive activities of *X. hominickii* [12]. The loss of Lrp function facilitated bacterial growth. Thus, there is a trade-off between bacterial growth and defense against the host insect’s immune response by up-regulation of *Lrp* expression at the early stages of bacterial infection [39]. Along with Lrp, two other transcriptional factors, CpxR and LrhA, also affect the infection and growth of *X. nematophila*, with the temporal regulation of their expression facilitating the progression from infection to reproduction in bacterial growth phases [40]. In the susceptible host, *M. vitrata*, bacterial growth rate might be explained by differential expression control of regulon subsets among the six *Xenorhabdus* strains. Based on this trade-off hypothesis, the relatively rapid growth rate of *Xenorhabdus* bacteria in the tolerant *T. molitor* could be due to the suppression of *Lrp* expression by unknown host factors as mentioned previously. The suppression of *Lrp* expression would inhibit the biosynthesis of secondary metabolites responsible for inducing host immunosuppression.

Regulation of *NRPS* expression was functionally linked to the virulence variation observed in the six *Xenorhabdus* strains. These strains produced secondary metabolites classified into categories such as peptides, indoles, hydrocarbons, phenylamides, and others. Mollah and Kim [41] analyzed secondary metabolites from 14 different strains of *Photorhabdus* and *Xenorhabdus*, identifying 70 potent virulent compounds, including 2-ethyl-1-hexanol, a metabolite specific to *Xenorhabdus*, which was detected in our current analysis in the culture broth of *X. nematophila* (see Appendix A). Unlike the previous study, our study revealed several peptides. Although both studies used GC-MS for chemical analysis, the extraction methods differed in the organic solvents used: hexane in the previous study and ethyl acetate in the current assay. Ethyl acetate’s less nonpolar nature compared to hexane likely facilitates easier extraction of peptides. cPF peptides were predominantly produced in the virulent strains of each bacterial species. Genome analysis of the bacteria revealed that all three species encode seven to eight NRPS genes. However, these were not co-linear in our synteny analysis, indicating significant gene shuffling during speciation. In this evolutionary divergence, molecular interactions between the bacteria and their insect hosts likely play a crucial role in shaping genome size and composition. For example, two strains of *X. bovienii* displayed different genome sizes and gene compositions, with several virulence genes becoming pseudogenes in the susceptible strain [42].

A peptide metabolite, cPF, was identified as a virulent factor of *Xenorhabdus*, significantly influencing virulence variation. While further investigation is required to identify the specific NRPS responsible for cPF production, NRPS6 of *X. hominickii* is likely responsible for its biosynthesis [12]. cPF inhibited both PLA_2_ activity and PO activity in our assays. PO activity relies on prostaglandin (PG) mediation among eicosanoids [43]. Therefore, cPF may indirectly inhibit PO by blocking PLA_2_, which in turn hampers PG biosynthesis. In a previous study, the expression of *Xh-NRPS6* was dependent on *Lrp* expression, which subsequently controlled cPF production in *X. hominickii* [44]. Another study showed that two virulent strains against *S. exigua* exhibited differential *Lrp* expression levels, correlating with variations in *NRPS* expression levels in *X. hominickii* [7]. These findings suggest that bacterial variation in insecticidal activity in this study can be attributed to the differential expression of the *Lrp* gene in at least the susceptible host, *M. vitrata*, since the addition of cPF enhanced bacterial virulence. Conversely, the relatively low expression of most *NRPS* genes in *T. molitor* might lead to inadequate cPF production by the bacteria, which fails to effectively inhibit the host immune responses.

Altogether, these results demonstrate that bacterial virulence in *Xenorhabdus* depends on immunosuppression of the host insects. Various secondary metabolites from *Xenorhabdus*, including NRPS-derived peptides, are crucial in suppressing immune responses by inhibiting PLA_2_. Consequently, the regulation of *NRPS* expression significantly influences bacterial pathogenicity against insect hosts.

## Figures and Tables

**Figure 1 insects-15-00710-f001:**
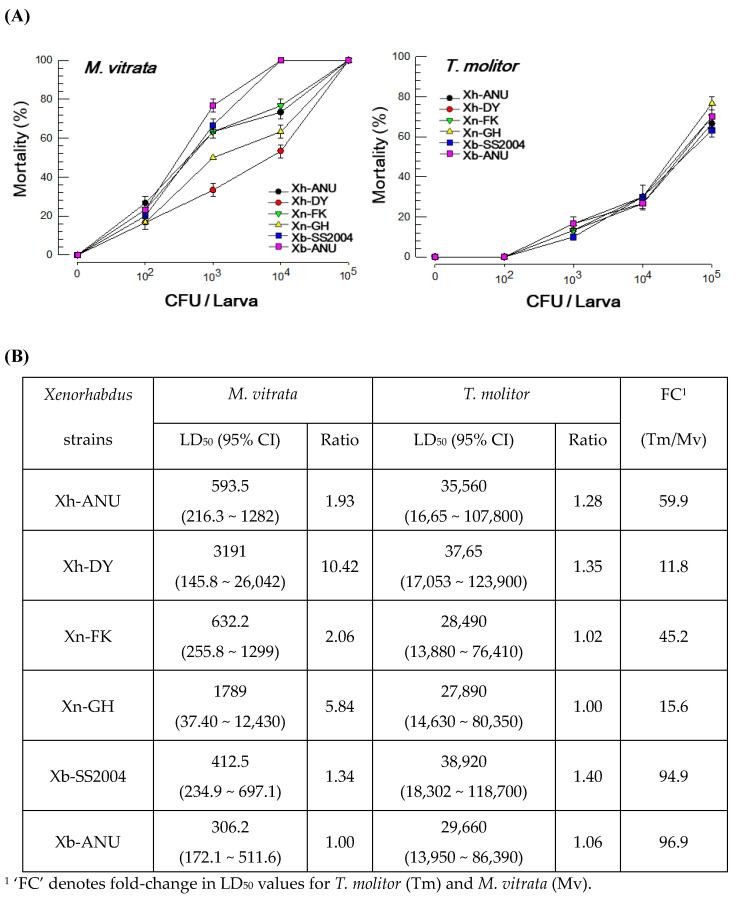
Variation in the insecticidal activities of six different *Xenorhabdus* strains against two distinct insect larvae: *Maruca vitrata* and *Tenebrio molitor*. The six *Xenorhabdus* strains are divided into two strains each from three bacterial species: *X. nematophila* (‘Xn’), *X. hominickii* (‘Xh’), and *X. bovienii* (‘Xb’). (**A**) Dose-mortality by hemocoelic injection of the bacteria. Mortality was measured at 3 days post-treatment. Each experimental unit included 10 larvae and was repeated three times with different cohorts. (**B**) Comparison of median lethal dose (LD_50_) and 95% confidence interval estimates among the six strains.

**Figure 2 insects-15-00710-f002:**
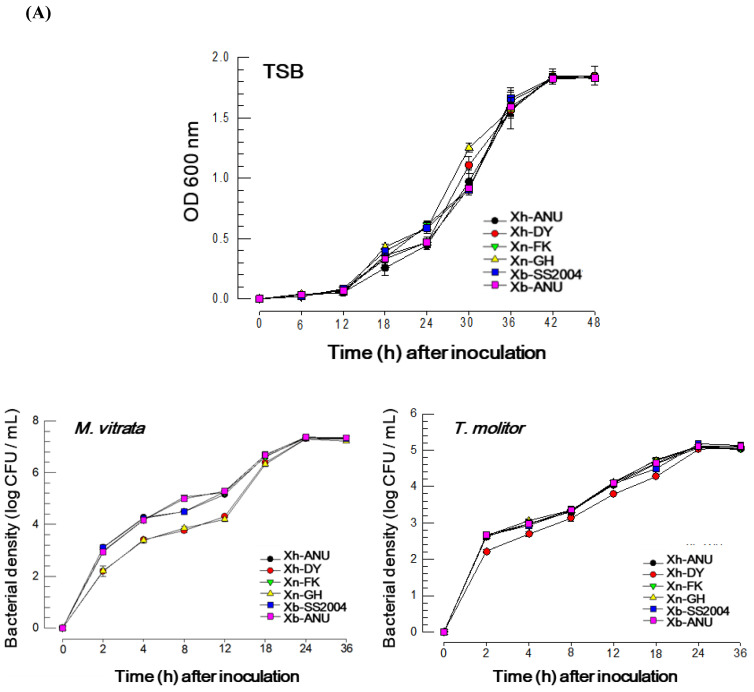
Influence of the bacterial growth rate and host immunosuppressive activity on the virulence of six different *Xenorhabdus* strains against two different insect larvae of *Maruca vitrata* and *Tenebrio molitor*. The six *Xenorhabdus* strains included two strains from each of three bacterial species: *X. nematophila* (‘Xn’), *X. hominickii* (‘Xh’), and *X. bovienii* (‘Xb’). (**A**) Bacterial growth profiles of the six different bacterial strains under various culture conditions. Bacterial growth (measured by OD600) in 1 L of TSB medium with an initial inoculation of 8 × 10^8^ CFU/L. Bacterial growth (measured by CFU) within the larval hemocoel of *M. vitrata* and *T. molitor* following an injection of 10^3^ CFU/larva. Each treatment was replicated three times independently. (**B**) Immunosuppressive activities of organic extracts from the six bacterial strains against PLA_2_ enzyme of *M. vitrata* and *T. molitor*. Enzyme activities were measured in sPLA_2_ and iPLA_2_ using various substrates. Each treatment was replicated three times independently. (**C**) Correlation analysis among four parameters: bacterial growth rate, sPLA_2_ inhibitory activity, iPLA_2_ inhibitory activity, and bacterial virulence. The Pearson correlation coefficient was calculated using Prism 10 (GraphPad). The bacterial growth rate is represented by the *r* values, two PLA_2_ inhibitory activities by the median inhibitory dose (IC_50_) values, and the virulence by the bacterial LD_50_ values.

**Figure 3 insects-15-00710-f003:**
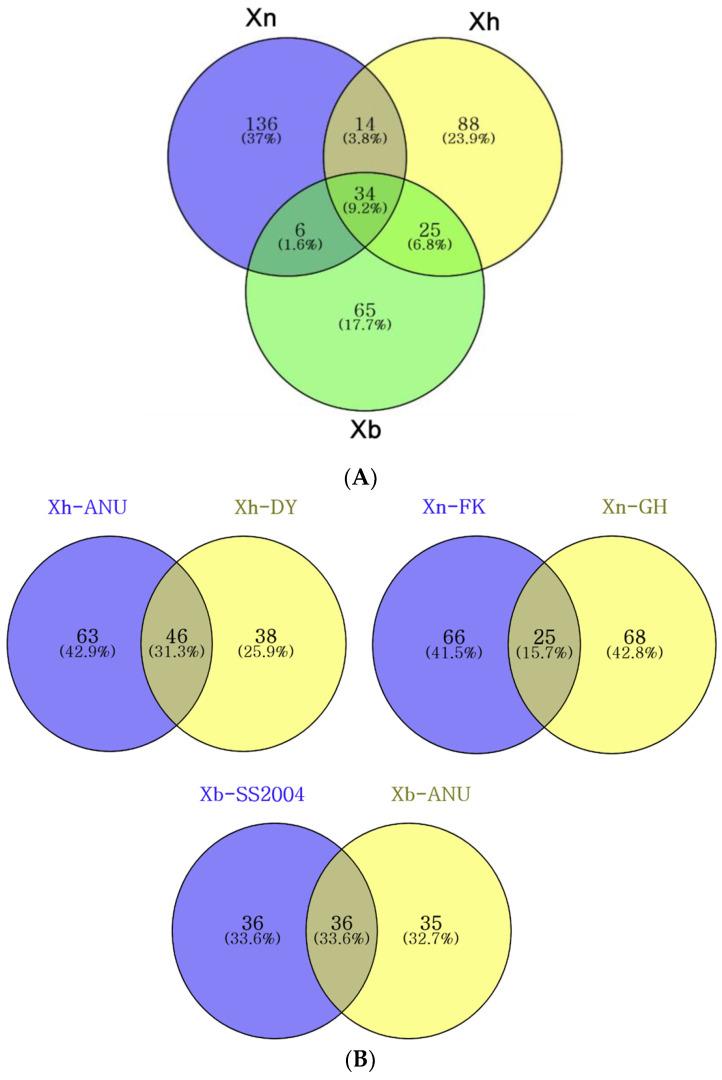
Comparison of the number of secondary metabolites extracted from the bacterial cultures among six different *Xenorhabdus* strains. The compounds were analyzed using GC-MS. These six *Xenorhabdus* strains included two strains each from three bacterial species: *X. nematophila* (‘Xn’), *X. hominickii* (‘Xh’), and *X. bovienii* (‘Xb’). (**A**) A Venn diagram comparing the numbers of secondary metabolites among the bacterial species. (**B**) Venn diagram comparisons between two strains within each of the three species, showing the numbers of secondary metabolites.

**Figure 4 insects-15-00710-f004:**
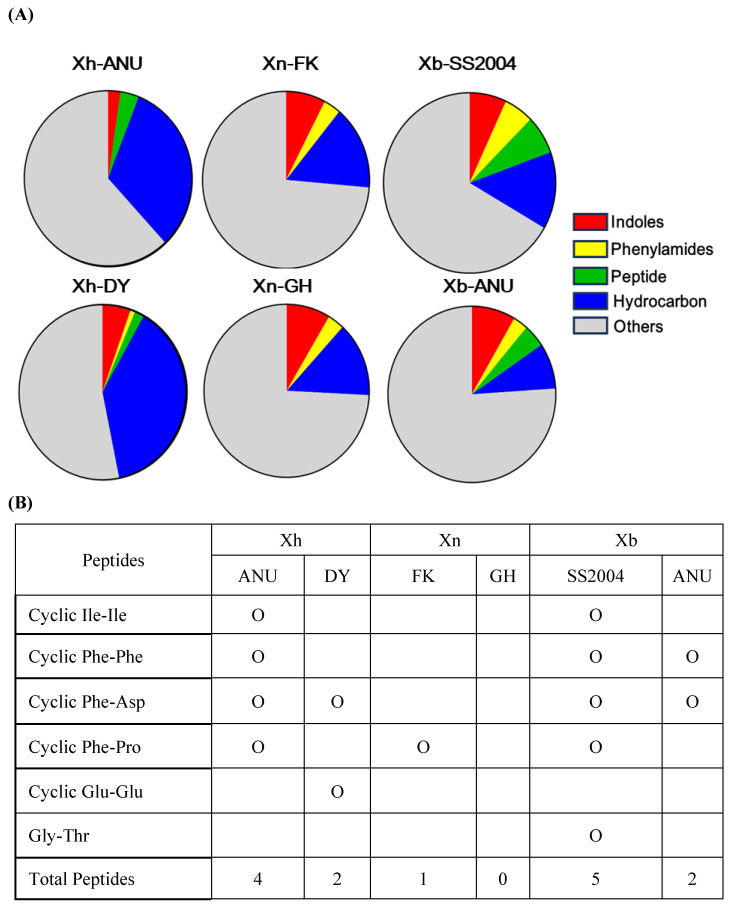
Comparison of the composition of secondary metabolites extracted from the bacterial cultures among six different *Xenorhabdus* strains. GC-MS analysis predicted these compounds and classified them into five groups: indoles, phenylamides, peptides, hydrocarbons, and others. The six *Xenorhabdus* strains each included two strains from three bacterial species: *X. nematophila* (‘Xn’), *X. hominickii* (‘Xh’), and *X. bovienii* (‘Xb’). (**A**) Comparison of the relative proportions of the five compound groups across the six *Xenorhabdus* strains. (**B**) Comparison of peptides across the six *Xenorhabdus* strains, with ‘O’ indicating the presence of the peptide.

**Figure 5 insects-15-00710-f005:**
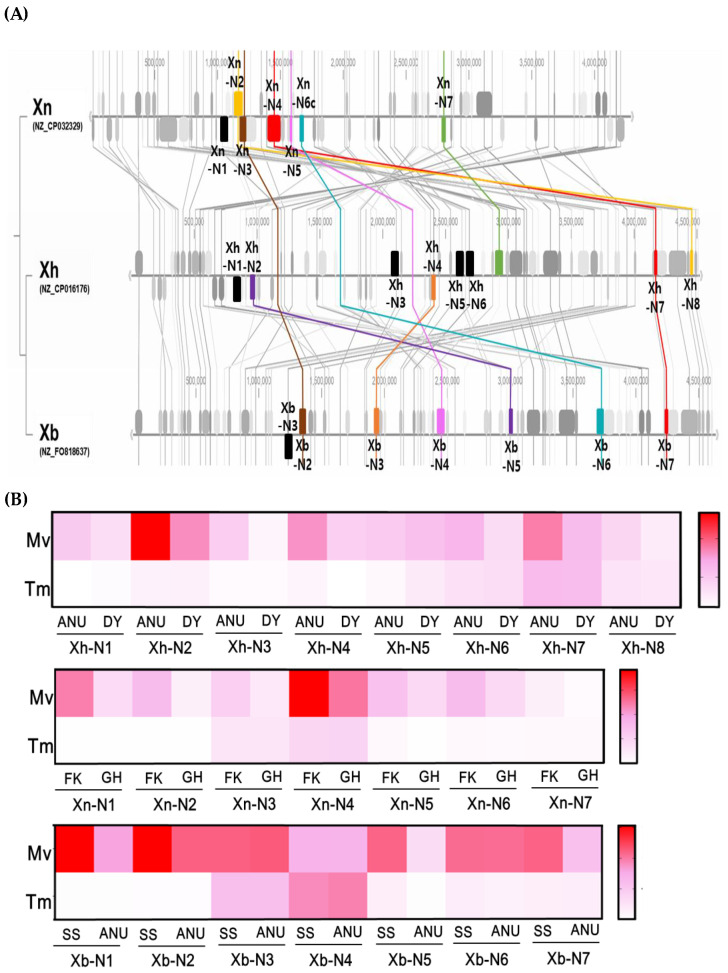
Genome mapping and expression profiles of various NRPS genes from different *Xenorhabdus* strains. Six *Xenorhabdus* strains comprising two strains from each of three bacterial species: *X. nematophila* (‘Xn’), *X. hominickii* (‘Xh’), and *X. bovienii* (‘Xb’). (**A**) Synteny analysis of different NRPS genes: seven NRPS genes (*Xn-N1*~*Xn-N7*) in Xn, eight NRPS genes (*Xh-N1*~*Xh-N8*) in Xh, and seven NRPS genes (*Xb-N1*~*Xb-N7*). GenBank accession numbers are provided in parentheses. (**B**) Comparative analysis of NRPS expression patterns in two strains from each *Xenorhabdus* species within two different insect hosts: *Maruca vitrata* (‘Mv’) and *Tenebrio molitor* (‘Tm’). Following injection of each bacterial strain into larvae at a concentration of 10^3^ CFU/larva, RT-qPCR was performed 12 h post-injection. Different red colors indicate the expression intensity.

**Figure 6 insects-15-00710-f006:**
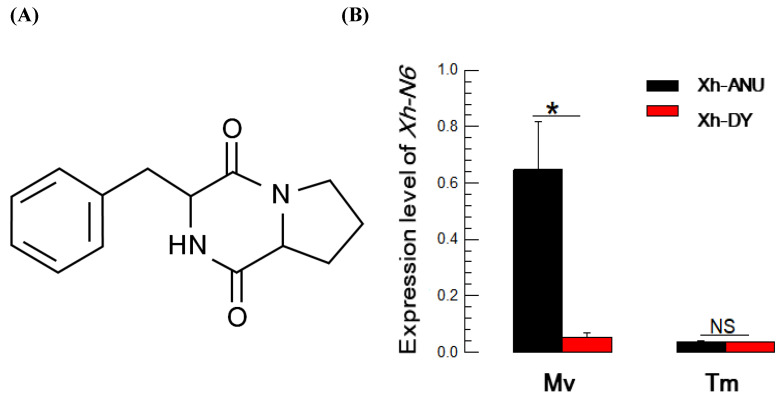
Immunosuppressive activity of cyclic Pro-Phe (cPF) compound against two different insect hosts: *Maruca vitrata* (‘Mv’) and *Tenebrio molitor* (‘Tm’). (**A**) Chemical structure. (**B**) Differential expression levels of *Xh-N6* genes in two strains of *Xenorhabdus hominickii* in the two different insect hosts. After injecting each bacterial strain into the larvae at a concentration of 10^3^ CFU/larva, RT-qPCR was performed 12 h post-injection. Each treatment was independently replicated three times. An asterisk indicates significant differences between the two strains, while ‘NS’ denotes no significant difference at Type I error = 0.05 (LSD test). (**C**) Inhibitory activity of cPF against two types of PLA_2_ activities (sPLA_2_ and iPLA_2_) in the enzyme extracts of the two insect hosts. The upper and lower case letters indicate the independent statistical analysis in each species. (**D**) The inhibitory activity of cPF against phenoloxidase (PO) activity in the hemolymph of both insect hosts. The treated concentration of cPF corresponds to the injected amount per larva. (**E**) Enhanced virulence of *X. hominickii* by the addition of cPF to the Xh-DY strain against Mv (left panel) or to the Xh-ANU strain against Tm. To assess this additive effect of cPF on virulence, 50 µg of the compound was co-injected with 10^3^ CFU of bacteria into the test larvae. Different letters above the standard deviation bars signify significant differences among means at Type I error = 0.05 (LSD test).

## Data Availability

The data from this study are available in the article.

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
