# Peer review of "A Comparative Analysis of Different *Xenorhabdus* Strains Reveals a Virulent Factor, Cyclic Pro-Phe, Using a Differential Expression Profile Analysis of Non-Ribosomal Peptide Synthetases"

_insects, 2024, doi:10.3390/insects15090710_

Round 1
Reviewer 1 Report
Comments and Suggestions for Authors
This is a very interesting study about the variability in pathogenic effects of Xenorhabdus on insects. The study was well conducted, and the MS was well written. I just have a few comments:
1. In all figures and tables: Please do not abbreviate scientific names.
2. Line 102: Please briefly describe the method to rear these insects.
3. Line 132: Please provide more information (e.g., temperature) about the bioassays.
4. 2.7 and 2.8 can be combined.
Overally, it is a very interesting and important study that can be published in Insects.
Author Response
Comment #1-1: In all figures and tables: Please do not abbreviate scientific names.
Response: Due to limited space in figures, the scientific names are abbreviated and the acronym are explained in the figure captions. All the scientific names in the figure captions are full-named at first.
Comment #1-2: 2. Line 102: Please briefly describe the method to rear these insects.
Response: Detailed conditions are added as follows:
2.1. Insect rearing
Larvae of M. vitrata were collected from adzuki bean (Vigna angularis) fields in Suwon, Korea. They were subsequently cultured on an artificial diet as outlined by Jung et al. [18]. Adults were provided with 10% sucrose. Larvae and adults of T. molitor were obtained from Bio Utility, Inc. (Andong, Korea) and reared on a wheat bran-based diet supple-mented with cabbage, following the methodology detailed by Liu et al. [19]. Rearing conditions were 25 ± 2oC temperature, 16:8 h (L:D) photoperiod, and 60 ± 10% relative humidity.
Comment #1-3: 3. Line 132: Please provide more information (e.g., temperature) about the bioassays.
Response: Detailed information including the test conditions are added as follows:
2.5. Bioassay on insecticidal activity of the bacteria
Third instar (L3) larvae of M. vitrata and L8 larvae of T. molitor were used. Freshly cultured bacteria were injected into each larva with 1 μL volume containing 1.5 × 10³ col-ony-forming units (CFU) using a micro-syringe. The treated larvae were kept at the rearing conditions with fresh diets. Mortality was assessed 72 h post-injection (pi) due to most dead insects were observed at 48 h pi. Each treatment included 10 larvae and was repli-cated three times with different cohorts. Control used an injection of PBS used for resus-pending the test bacteria.
Comment #1-4: 4. 2.7 and 2.8 can be combined.
Response: These two are combined.
Reviewer 2 Report
Comments and Suggestions for Authors
The manuscript entitled "A comparative analysis of different Xenorhabdus strains reveals a virulent factor, cyclic Pro-Phe, using a differential expression profile analysis of non-ribosomal peptide synthetases" by Jin et al., investigated different Xenorhabdus strains that exhibit varying levels of insecticidal virulence, influenced by growth rates and suppressive activities against host immune functions. The manuscript entitled "A comparative analysis of different Xenorhabdus strains reveals a virulent factor, cyclic Pro-Phe, using a differential expression profile analysis of non-ribosomal peptide synthetases" by Jin et al., investigated different Xenorhabdus strains that exhibit varying levels of insecticidal virulence, influenced by growth rates and suppressive activities against host immune functions. The authors demonstrated that virulent strains express non-ribosomal peptide synthetases (NRPS). Additionally, a peptide metabolite, cPF, was identified as a significant virulent factor of Xenorhabdus, influencing virulence variation. The manuscript's main premise is well-supported, but there are some unexplored observations and concerns about the controls used.
Major comments:
1) An overall comment is that several measurements used Third instar (L3) larvae of M. vitrata and L8 larvae of T. molitor except Phenoloxidase (PO) enzyme assay, where the authors used L5 stage. Can the authors provide more explanation about the choice of the larval stage in each experiment?
2) It would be beneficial to address a couple of concerns regarding the mortality assays. Firstly, including a negative control to demonstrate normal mortality in the absence of the virulent bacteria would strengthen the experiment. Additionally, specifying the medium used to suspend the bacteria inoculum would provide important clarity.
3) Nodulation is a cellular immune response that is dependent on PLA2 activity. Does Xenorhabdus strains infection induce or repress this mechanism?
Author Response
Comment #2-1: An overall comment is that several measurements used Third instar (L3) larvae of M. vitrata and L8 larvae of T. molitor except Phenoloxidase (PO) enzyme assay, where the authors used L5 stage. Can the authors provide more explanation about the choice of the larval stage in each experiment?
Response: The same instars were used in all the assays in each insect species. Thus, there is a correction in the PO activity as follows: “Hemolymph was collected from the test larvae used for bioassay and subsequently divided into hemocytes and plasma, adhering to the outlined protocol.”
Comment #2-2: It would be beneficial to address a couple of concerns regarding the mortality assays. Firstly, including a negative control to demonstrate normal mortality in the absence of the virulent bacteria would strengthen the experiment. Additionally, specifying the medium used to suspend the bacteria inoculum would provide important clarity.
Response: Detailed information including the test conditions are added as follows:
2.5. Bioassay on insecticidal activity of the bacteria
Third instar (L3) larvae of M. vitrata and L8 larvae of T. molitor were used. Freshly cultured bacteria were injected into each larva with 1 μL volume containing 1.5 × 10³ col-ony-forming units (CFU) using a micro-syringe. The treated larvae were kept at the rearing conditions with fresh diets. Mortality was assessed 72 h post-injection (pi) due to most dead insects were observed at 48 h pi. Each treatment included 10 larvae and was repli-cated three times with different cohorts. Control used an injection of PBS used for resus-pending the test bacteria.
Comment #2-3: Nodulation is a cellular immune response that is dependent on PLA2 activity. Does Xenorhabdus strains infection induce or repress this mechanism?
Response: Even though nodulation assay was not used in this study, it is dependent on PLA2 activity. Thus the bacterial strains inhibits the cellular immune response.
Reviewer 3 Report
Comments and Suggestions for Authors
Xenorhabdus is well-known for its association with the toxicity of entomopathogenic nematodes. Previous studies have demonstrated variability in insecticidal virulence among different strains. However, the underlying mechanisms remain unclear. In this study, the authors first evaluated the virulence variation among six Xenorhabdus bacterial strains and then investigated the potential role of peptide metabolites in this variation. Overall, the study is well-executed, with sufficient data and a logical structure. I have only a few minor suggestions, which are listed below:
- Line 92: The rationale for using GC-MS with ethyl acetate as a solvent should be explained.
- Line 91: A brief introduction to the two insect species would be helpful, such as whether they are considered pests.
- Lines 99-103: A brief description of the methodology for rearing the two insect species should be provided, even though there are references. This issue should be addressed throughout the M&M section.
- Line 133: What was the total volume of the bacteria-containing solution injected into the insects? The experimental description is too vague.
- Line 134: Why did the authors record the mortality of both insect species at the same time point post-injection? The two insects belong to different orders and have distinct immune systems. Using the same method for assaying toxicity may not be appropriate.
- Line 137: Is 2 μL too much for T. molitor? Did the study include a blank control?
- Lines 218-226: The Latin names of the two species should be italicized (correct this throughout the manuscript). Additionally, how was the LD50 determined? No experimental details are provided in the M&M section.
- Line 246: This is a bacterial growth curve—did the authors use a one-way ANOVA for comparison? If so, how was it applied?
- Lines 396-400: The focus of this study is not on identifying new compounds but rather on exploring the reasons for the virulence differences among strains. This section seems off-topic and should be revised accordingly.
Author Response
Comment #3-1: Line 92: The rationale for using GC-MS with ethyl acetate as a solvent should be explained.
Response: Following information is added: “Secondary metabolites from the six strains were analyzed using GC-MS on ethyl acetate extracts from the bacterial culture broth containing PLA2 inhibitors [9].”
Comment #3-2: Line 91: A brief introduction to the two insect species would be helpful, such as whether they are considered pests.
Response: Following is added: “a lepidopteran Maruca vitrata and a coleopteran Tenebrio molitor, which have been demon-strated in their immune responses controlled by eicosanoids [7].”
Comment #3-3: Lines 99-103: A brief description of the methodology for rearing the two insect species should be provided, even though there are references. This issue should be addressed throughout the M&M section.
Response: Detailed information including the test conditions are added as follows:
2.5. Bioassay on insecticidal activity of the bacteria
Third instar (L3) larvae of M. vitrata and L8 larvae of T. molitor were used. Freshly cultured bacteria were injected into each larva with 1 μL volume containing 1.5 × 10³ col-ony-forming units (CFU) using a micro-syringe. The treated larvae were kept at the rearing conditions with fresh diets. Mortality was assessed 72 h post-injection (pi) due to most dead insects were observed at 48 h pi. Each treatment included 10 larvae and was repli-cated three times with different cohorts. Control used an injection of PBS used for resus-pending the test bacteria.
Comment #3-4: Line 133: What was the total volume of the bacteria-containing solution injected into the insects? The experimental description is too vague.
Response: See the response to Comment #3.3.
Comment #3-5: Line 134: Why did the authors record the mortality of both insect species at the same time point post-injection? The two insects belong to different orders and have distinct immune systems. Using the same method for assaying toxicity may not be appropriate.
Response: The bacteria killed the insects at 48-72 h in both. See the the response to Comment #3.3.
Comment #3-6: Line 137: Is 2 μL too much for T. molitor? Did the study include a blank control?
Response: For this study, we used the bigger size of larvae were used and revised the text as follows: “In this assay, we used the last instar larvae of M. vitrata and the matured larvae (> 10 mm body length) of T. molitor.”
Comment #3-7: Lines 218-226: The Latin names of the two species should be italicized (correct this throughout the manuscript). Additionally, how was the LD50 determined? No experimental details are provided in the M&M section.
Response: Scientific names are italic. Parameters including LD50 are described in section 2.11.
Comment #3-8: Lines 396-400: The focus of this study is not on identifying new compounds but rather on exploring the reasons for the virulence differences among strains. This section seems off-topic and should be revised accordingly.
Response: Rephrased the paragraph as follows: “This study provides information concerning the secondary metabolites produced by var-ious strains of Xenorhabdus bacteria in terms of compounds and biological activities to ex-plain the virulence variation among different strains of Xenorhabdus.”